# Melatonin Promotes Iron Reactivation and Reutilization in Peach Plants under Iron Deficiency

**DOI:** 10.3390/ijms242216133

**Published:** 2023-11-09

**Authors:** Lijin Lin, Zhiyu Li, Caifang Wu, Yaxin Xu, Jin Wang, Xiulan Lv, Hui Xia, Dong Liang, Zhi Huang, Yi Tang

**Affiliations:** 1College of Horticulture, Sichuan Agricultural University, Chengdu 611130, China; 2021205022@stu.sicau.edu.cn (Z.L.); 810016@hainanu.edu.cn (C.W.); 2021305069@stu.sicau.edu.cn (Y.X.); 14224@sicau.edu.cn (J.W.); 10998@sicau.edu.cn (X.L.); xiahui@sicau.edu.cn (H.X.); huangzhi@sicau.edu.cn (Z.H.); tangyi@sicau.edu.cn (Y.T.); 2Yazhou College, Hainan University, Sanya 570228, China

**Keywords:** iron deficiency, melatonin, peach, transcriptome profiling

## Abstract

The yellowing of leaves due to iron deficiency is a prevalent issue in peach production. Although the capacity of exogenous melatonin (MT) to promote iron uptake in peach plants has been demonstrated, its underlying mechanism remains ambiguous. This investigation was carried out to further study the effects of exogenous MT on the iron absorption and transport mechanisms of peach (*Prunus persica*) plants under iron-deficient conditions through transcriptome sequencing. Under both iron-deficient and iron-supplied conditions, MT increased the content of photosynthetic pigments in peach leaves and decreased the concentrations of pectin, hemicellulose, cell wall iron, pectin iron, and hemicellulose iron in peach plants to a certain extent. These effects stemmed from the inhibitory effect of MT on the polygalacturonase (PG), cellulase (Cx), phenylalanine ammonia-lyase (PAL), and cinnamoyl-coenzyme A reductase (CCR) activities, as well as the promotional effect of MT on the cinnamic acid-4-hydroxylase (C4H) activity, facilitating the reactivation of cell wall component iron. Additionally, MT increased the ferric-chelate reductase (FCR) activity and the contents of total and active iron in various organs of peach plants under iron-deficient and iron-supplied conditions. Transcriptome analysis revealed that the differentially expressed genes (DEGs) linked to iron metabolism in MT-treated peach plants were primarily enriched in the aminoacyl-tRNA biosynthesis pathway under iron-deficient conditions. Furthermore, MT influenced the expression levels of these DEGs, regulating cell wall metabolism, lignin metabolism, and iron translocation within peach plants. Overall, the application of exogenous MT promotes the reactivation and reutilization of iron in peach plants.

## 1. Introduction

Iron plays an important role in various plant physiological processes, including chlorophyll precursor synthesis and chloroplast formation [1]. Iron deficiency impedes the formation of pyrrole and porphyrin rings, thus obstructing chlorophyll synthesis [1,2]. This phenomenon leads to pronounced changes in the chloroplast lamellar structure, leading to thylakoid arrangement disruption, chloroplast destabilization, and diminished chlorophyll content [3]. Additionally, iron deficiency decreases the cytochrome oxidase activity, the content of iron-containing compounds, and the iron-dependent enzyme activity. This disruption impairs the electron transport chain during photosynthesis, thereby impeding the photosynthetic processes [4,5]. Furthermore, the manifestation of leaf yellowing resulting from iron deficiency exerts a pronounced influence on plant growth and development, consequently inducing substantial losses in both yield and quality [6].

Two distinct uptake strategies are employed by higher plants to acquire less-bioactive iron from the soil: (1) Mechanism I, grounded in a reduction strategy, is applicable to dicotyledonous plants and non-grass monocotyledonous plants. Under iron-deficient conditions, the root plasma membrane’s H^+^-ATPase releases protons, increasing the activity of ferric-chelate reductase (FCR) to convert Fe^3+^ to Fe^2+^, facilitating its uptake by plant roots [7,8,9]. (2) Mechanism II, rooted in a chelation strategy, is tailored to monocotyledonous gramineous plants. Here, Fe^3+^ is chelated upon the release of a phytoferric carrier in response to iron deficiency. This chelate is subsequently transported intracellularly by the transporter protein YS1 and its cognate protein YSL, located in the cytoplasmic membranes of roots. Once inside, Fe^3+^ binds to either nicotinamide (stored in root cytosol vesicles) or citric acid (for xylem-to-above-ground transport). These iron complexes are transported upward to the leaf plasma membrane, where they are then reduced by FCR within the leaves [10,11,12].

The absorption of iron by plants depends not only on the quantity of available iron surrounding the roots but also on the influence of specific genes related to iron absorption. These genes include *ferric reduction oxidase 7* (*FRO7*), *vacuolar iron transporter 1* (*VIT1*), *natural resistance-associated macrophage protein 3* (*NRAMP3*), *ferric reduction oxidase 4* (*FRO4*), *vacuolar iron transporter homolog 4* (*VITH4*), etc. [13,14,15]. Additionally, exogenous phytohormones like ethylene and auxin play a role in iron uptake regulation [16,17], suggesting their potential in controlling plant iron uptake. Melatonin (MT), an indole derivative of tryptophan, regulates a plethora of physiological functions in plants [18,19]. It has a role in governing plant growth, development, germination, and leaf senescence delay [20,21]. Furthermore, MT boasts a reputation as a potent endogenous free-radical scavenger with antioxidant properties, contributing to the management of the metabolic equilibrium of intracellular reactive oxygen species [20,21,22].

Previous studies have shown that exogenous MT promotes iron uptake in fruit trees under normal conditions [23,24]. Under iron-deficient conditions, the application of MT on *Arabidopsis* treats iron-deficiency-induced leaf yellowing by inducing polyamine-mediated nitric oxide (NO) accumulation and influencing the re-migration of cell wall iron [3]. In cucumber, exogenous MT not only elevates the endogenous MT content but also augments the photosynthesis rate, antioxidant enzyme activity, and secondary metabolism-related enzyme activities. This response aids in addressing the iron deficiency and upregulates the transcript levels of *FRO2* and *IRT1* to facilitate iron absorption [25]. These findings suggest that exogenous MT can enhance the iron uptake in horticultural crops under iron-deficient conditions. However, the precise regulatory mechanism through which MT impacts the iron metabolism in horticultural crops remains elusive.

Peach (*Prunus persica*) holds significance as a major fruit globally [26]. Nevertheless, the escalating occurrence of yellowing in peach leaves due to soil alkalinity in numerous regions has adversely affected its fruit yield and quality [27]. In a previous study, we established that exogenous MT could promote iron uptake in peach plants, with significant effects observed at 100–150 µmol/L MT concentrations [24]. Nonetheless, the mechanism underlying this phenomenon remains elusive. To address this gap, this investigation was carried out to further study the effects of exogenous MT (at a concentration of 100 µmol/L) on the iron absorption and transport mechanisms of peach plants under iron-deficient conditions through transcriptome sequencing. The aim of this study was to provide insights for mitigating iron deficiency-induced yellowing in peach plants.

## 2. Results

### 2.1. Photosynthetic Pigment Content and Leaf Color of Peach Seedlings

Iron deficiency leads to decreased contents of chlorophyll *a*, chlorophyll *b*, and carotenoids in both the upper and lower leaves of peach seedlings (Figure 1A). In the lower leaves, MT+Fe treatment increased the contents of chlorophyll *a* and chlorophyll *b*, with no significant impact on carotenoid content compared to Fe treatment. Similarly, MT−Fe treatment increased the contents of all three pigments in lower leaves compared to −Fe treatment. In the upper leaves, MT+Fe treatment only increased chlorophyll *a* content compared to Fe treatment, while MT−Fe treatment only increased chlorophyll *a* content compared to −Fe treatment. Notably, MT treatment showed no significant effects on the contents of chlorophyll *b* and carotenoid in the upper leaves under both iron deficiency and iron supply conditions. Moreover, MT treatment deepened the green color intensity of upper leaves under both iron deficiency and iron supply (Figure 1B) because MT increased the chlorophyll *a* content.

### 2.2. Pectin, Hemicellulose, and Their Iron Contents in Peach Seedlings

Iron deficiency prompted an increase in the pectin and hemicellulose contents of various organs of peach seedlings (Figure 2A,B). MT treatments resulted in decreased pectin and hemicellulose contents in different organs under both iron deficiency and iron supply conditions. Additionally, iron deficiency increased the contents of pectin iron, hemicellulose iron, and cell wall iron in various organs compared to iron supply (Figure 3A–C). MT+Fe treatment decreased the pectin iron content in roots, stems, and upper leaves compared to Fe treatment to some extent. Similarly, MT−Fe treatment decreased the pectin iron content in the roots and lower leaves compared to −Fe treatment. MT+Fe treatment increased the hemicellulose iron content in stems, while exhibiting no significant impact on the roots, lower leaves, and upper leaves compared to Fe treatment. Conversely, MT−Fe treatment reduced the hemicellulose iron content in roots, raised it in stems, and had no significant effect on lower leaves and upper leaves compared to −Fe treatment. MT+Fe treatment decreased the cell wall iron content in the stems, lower leaves, and upper leaves, while showing no significant change in the roots compared to Fe treatment. Similarly, MT−Fe treatment decreased the cell wall iron content in the roots, stems, and lower leaves, while having no significant impact on the upper leaves compared to −Fe treatment.

### 2.3. Total Iron and Active Iron Content in Peach Seedlings

Under iron deficiency conditions, the contents of total iron and active iron in various organs of peach seedlings were somewhat lower compared to those in conditions of iron supply (Figure 4A,B). Conversely, under iron supply conditions, MT treatment increased the total iron content in stems and the active iron content in roots and stems. However, MT treatment showed no significant effects on the contents of total iron and active iron in other organs. Under iron deficiency conditions, MT treatment increased the total iron content in the roots, lower leaves, and upper leaves, as well as the active iron content in the stems, lower leaves, and upper leaves. Notably, for MT−Fe treatment, there were no significant effects on the total iron content in stems and the active iron content in roots compared to −Fe treatment. In addition, iron deficiency reduced the proportion of active iron content to total iron content in various organs compared to iron supply (Figure 4C). MT+Fe treatment only increased the proportion of active iron content to total iron content in the roots, while it had no significant effects on that in other organs compared to Fe treatment. MT−Fe treatment increased the proportion of active iron content to total iron content in the stems and upper leaves, while it had no significant effects on that in the roots and lower leaves compared to −Fe treatment.

### 2.4. Enzyme Activities: PG, Cx, FCR, and Lignin-Metabolism-Related Enzymes in Peach Seedlings

Iron deficiency increased the activities of PG and Cx in peach seedlings (Figure 5A). Conversely, MT treatment reduced the activities of both PG and Cx, regardless of iron deficiency or iron supply conditions. The order of PG and Cx activities was observed as follows: −Fe > MT−Fe > Fe > MT+Fe. Furthermore, iron deficiency increased FCR activity and decreased PAL activity to a certain degree compared to iron supply (Figure 5B). Under conditions of both iron deficiency and iron supply, MT treatment increased FCR activity while moderately decreasing PAL activity. Additionally, iron deficiency increased the activities of C4H, CCR, 4CL, and CAD compared to iron supply (Figure 5C,D). Under both iron deficiency and iron supply conditions, MT treatment increased the activities of C4H and CCR, while exhibiting no significant impact on the activities of 4CL and CAD.

### 2.5. DEGs following Treatment

Upon comparing different treatments, a total of 810 DEGs were identified in the −Fe vs. Fe comparison, including 396 upregulated and 414 downregulated genes (Figure 6A). In the MT−Fe vs. −Fe comparison, a total of 154 DEGs were observed, including 57 upregulated and 97 downregulated genes. For the MT+Fe vs. Fe comparison, a total of 149 DEGs were detected, including 89 upregulated and 60 downregulated genes. Moreover, the MT−Fe vs. MT+Fe comparison exhibited 408 DEGs, with 196 upregulated and 212 downregulated genes. Furthermore, the Venn diagram demonstrated that there were five shared DEGs among all treatment groups (Figure 6B).

### 2.6. Functional Classification of DEGs

By leveraging the COG, GO, KEGG, KOG, NR, Pfam, Swiss-Prot, and EggNOG databases, the corresponding Unigene annotation information underwent analysis via Blast 2.9.0 software (National Library of Medicine, Bethesda, MD, USA) comparison (Table 1). GO enrichment analyses revealed distinct patterns among the various comparisons. In −Fe vs. Fe, MT−Fe vs. −Fe, MT+Fe vs. Fe, and MT−Fe vs. MT+Fe, 560, 95, 81, and 282 DEGs were enriched, respectively. The GO annotation exhibited three primary branches encompassing biological process, molecular function, and cellular component categories (Figure 7). Notably, within the biological process branch, metabolic process, cellular process, and single-organism process subcategories displayed higher gene frequencies. Similarly, catalytic activity and binding subcategories prevailed within the molecular function branch, while the cell, cell part, and membrane subcategories were prominent within the cellular component branch. Subsequently, KEGG pathway enrichment analysis identified 338, 48, 57, and 153 DEGs enriched in −Fe vs. Fe, MT−Fe vs. −Fe, MT+Fe vs. Fe, and MT−Fe vs. MT+Fe, respectively (Table 1). In the −Fe vs. Fe comparison, 192 DEGs were annotated to 84 KEGG pathways, with 10 significantly enriched pathways (*p* < 0.05). These enriched pathways notably included ribosome biogenesis in eukaryotes, photosynthesis, and photosynthesis antenna proteins (Table 2, Figure 8). Of significance, for the −Fe vs. Fe comparison, ribosome biogenesis in eukaryotes featured 19 annotated DEGs (all downregulated), while photosynthesis antenna proteins and photosynthesis showcased 14 DEGs (13 upregulated and 1 downregulated), and 12 DEGs (all downregulated), respectively. For the MT−Fe vs. −Fe comparison, 31 DEGs were annotated to 38 KEGG pathways, with aminoacyl-tRNA biosynthesis as the sole significantly enriched pathway (*p* < 0.05) and featuring 3 annotated DEGs (2 upregulated and 1 downregulated). In MT+Fe vs. Fe, 39 DEGs were annotated to 25 KEGG pathways, with protein processing in the endoplasmic reticulum and monoterpenoid biosynthesis as significantly enriched pathways (*p* < 0.05). These pathways featured 11 DEGs (10 upregulated and 1 downregulated) and 3 DEGs (2 upregulated and 1 downregulated), respectively. Lastly, MT−Fe vs. MT+Fe comparison demonstrated 96 DEGs annotated to 69 KEGG pathways, with the sole significantly enriched pathway (*p* < 0.05) being predominantly associated with photosynthesis antenna proteins and photosynthesis. In this context, six DEGs (all upregulated) and five DEGs (all upregulated) were annotated, respectively.

### 2.7. Transcription Factor Analysis of DEGs

The examination of transcription factor families through expression profiling libraries in −Fe vs. Fe, MT−Fe vs. −Fe, MT+Fe vs. Fe, and MT−Fe vs. MT+Fe comparisons yielded a plethora of differential transcription factors (Table 3). These encompassed 17 diverse transcription factor families, such as Trihelix, TCP, RAP, RADIALIS, PIF, NAC, MYB, IFH, ICE1, HY5, HSP, GATA, ERF, bZIP, bHLH, AP2, and WRKY. Delving into these transcription factor families illuminated the prominence of certain families. The MYB (six members), WRKY (five members), HSP (four members), bHLH (three members), and ERF (three members) families exhibited notable representation. Intriguingly, members within these transcription factor families also participated in plant responses to abiotic stress.

### 2.8. qRT-PCR Analysis of the DEGs

To enhance the fidelity of the expression trends exhibited by the DEGs under −Fe, MT−Fe, and MT+Fe treatments, this study employed qRT-PCR analysis (Figure 9). The findings underscored a congruence between the expression trends of eight screened DEGs observed through both RPKM and qRT-PCR analyses. In particular, the qRT-PCR results indicated a consistent modulation of gene expression. Specifically, in comparison to conditions of iron deficiency or iron supply, MT treatment showed a downregulation in the relative expression levels of iron-uptake-metabolism-associated genes *VITH4* and *NRAMP3*, as well as the pectin-metabolism-related gene *PG*. Concurrently, there was an observed upregulation in the relative expression levels of iron-uptake-metabolism-related genes *FRO7* and *FRO4*, along with the lignin-metabolism-linked genes *ICS2*, *CCR1*, and *4CL*, to a discernible extent.

## 3. Discussion

Extensive study has demonstrated that exogenous MT can enhance photosynthesis, sustain cell membrane integrity, and prevent chlorophyll degradation [28]. Moreover, under stress conditions, MT inhibits the degradation of chlorophyll in plants [29]. In this study, MT increased the green color intensity of peach seedling leaves under both iron deficiency and iron supply conditions. Additionally, MT increased the content of photosynthetic pigments in peach seedlings during iron deficiency and iron supply conditions. These results are consistent with the previous studies [28,29], suggesting that MT could promote photosynthetic pigments’ synthesis under iron supply conditions and prevent their degradation in iron-deficient settings. This phenomenon could be attributed to MT’s role in regulating photosynthetic physiology [18,30], as well as its capacity to expedite reactive oxygen species scavenging during stress conditions [31,32].

Iron deficiency triggers plants to recycle plasmalemma exosomal iron (comprising about 75% of the total iron in roots) for reuse [33,34]. The cell wall, primarily composed of cellulose, pectin, and hemicellulose, constitutes the principal component of the root’s plasmic ectodomain [35]. This wall features numerous cation-binding sites capable of binding multiple metal cations, underscoring the critical role of root iron reutilization in plant resilience against iron deficiency stress [36,37]. Diminished PAL activity corresponds to a lowered lignin content in plants. Concurrently, CCR reduces the CoA esters of three hydroxycinnamic acids to their corresponding cinnamic aldehydes, which are subsequently reduced by CAD to form the respective cinnamic alcohols [38,39,40,41]. Previous studies indicate that MT heightens mineral element uptake in plants [42] and elevates soluble iron content in plant roots and leaves by stimulating cell wall iron reactivation [3]. In this study, MT decreased the contents of cell wall iron, pectin iron, and hemicellulose iron in peach seedlings by suppressing PG and Cx activities and also reduced pectin and hemicellulose levels under iron deficiency and iron supply conditions. MT also decreased the activities of PAL and CCR while increasing C4H activity in peach seedlings. No significant effect was observed on the activities of 4CL and CAD under iron supply and iron deficiency conditions. These results indicate that MT could reactivate cell wall iron, pectin iron, and hemicellulose iron in peach seedlings. Furthermore, these results also suggest MT’s capacity to regulate iron release from the cell wall components of peach seedling by modulating enzyme activities associated with the cell components under iron-deficient conditions.

The induction of FCR activity in the plant root protoplasmic membrane is often regarded as the limiting step in iron uptake for mechanism I plants [43]. FCR reduces Fe^3+^ to Fe^2+^ in the medium, facilitating the uptake of Fe^2+^ into the plant root cytoplasm [44]. Similarly, in peach seedlings, MT increases total and active iron contents by increasing FCR activity [24]. In this study, MT also increased the FCR activity of peach seedlings under iron supply conditions. However, MT increased the total iron content in the stems and the active iron content in the roots and stems of peach seedlings, without significantly affecting other organs under iron supply conditions. To a certain extent, these findings are consistent with the observations of Lin et al. (2022) [24]. The rationale behind this could be attributed to the ample Fe^2+^ absorption from the nutrient solution in this study, leading to a negligible MT effect. On the other hand, cultivation in mildly alkaline soil may result in Fe^2+^ deficiencies, rendering a more pronounced MT effect [24]. Furthermore, under iron deficiency conditions, MT also increased the FCR activity in peach seedlings, along with increasing the total and active iron contents in various organs to some extent. Moreover, MT increased the proportion of active iron content to total iron content in stems and upper leaves under an iron deficiency condition. These results are consistent with the findings of Lin et al. (2022) [24], suggesting that MT might indeed increase iron uptake and reutilization in plants under Fe^2+^ deficiency conditions.

The transcriptome analysis in this study identified 810, 154, 149, and 408 functionally annotated DEGs in −Fe vs. Fe, MT−Fe vs. −Fe, MT+Fe vs. Fe, and MT−Fe vs. MT+Fe, respectively. GO annotation of these DEGs revealed prevalent categories in biological processes, molecular functions, and cellular components, including metabolic process, cellular process, single-organism process, catalytic activity, binding, cell, cell part, and membrane. These outcomes imply that iron deficiency and MT treatment influence genetic information transfer and processing and cellular components in peach seedlings and significantly contribute to iron storage and reutilization within the cellular components. KEGG pathway enrichment analysis unveiled specific pathways affected by the treatments. Iron deficiency (−Fe) vs. Iron supply (Fe) was primarily linked to pathways related to ribosome biogenesis in eukaryotes, photosynthesis, and photosynthesis antenna proteins. Meanwhile, MT−Fe vs. −Fe exhibited an association with the aminoacyl-tRNA biosynthesis pathway. MT+Fe vs. Fe was predominantly connected to protein processing in the endoplasmic reticulum and monoterpenoid biosynthesis pathways. Lastly, MT−Fe vs. MT+Fe prominently involved pathways concerning photosynthesis antenna proteins and photosynthesis. These observations collectively underscore the intricate response mechanisms of peach seedlings to iron deficiency stress, involving multifaceted physiological processes and metabolic pathways. Notably, pathways linked to photosynthesis and photosynthesis antenna proteins displayed significant impact and enrichment with numerous DEGs.

Recent research has highlighted the involvement of certain transcription factors as pivotal genes in the iron deficiency response signaling pathway in plants. The foremost transcription factor recognized in mechanism I plants for regulating the iron deficiency response is FER in tomatoes. FER encodes a bHLH transcription factor and is believed to play a role in regulating iron uptake in roots. Additionally, four other bHLH transcription factors (bHLH038, bHLH039, bHLH100, and bHLH101) are also prominently induced by iron deficiency [45,46]. In this study, a comprehensive assessment of transcription factors, spanning 17 distinct families, such as Trihelix, TCP, RAP, RADIALIS, PIF, NAC, MYB, IFH, ICE1, HY5, HSP, GATA, ERF, bZIP, bHLH, AP2, and WRKY, was conducted for −Fe vs. Fe, MT−Fe vs. −Fe, MT+Fe vs. Fe, and MT−Fe vs. MT+Fe comparisons. Among these, the MYB, WRKY, HSP, bHLH, and ERF families demonstrated significant annotation coverage and are known to play roles in plant responses to iron stress [47].

Throughout the course of extensive evolutionary adaptation, plants have developed comprehensive physiological responses and intricate molecular regulatory mechanisms to contend with iron deficiency conditions [48]. Noteworthy studies have underscored the importance of vesicle-contained iron as a vital resource for iron recycling within plants. *NRAMP* can regulate the iron absorption and transportation. It is upregulated under iron deficiency stress to facilitate iron efflux from vesicles [14,15,49,50]. In *Arabidopsis*, *AtVIT1*, positioned in vesicular membranes, is primarily responsible for ferrying iron from the cytoplasm to the vesicles. *AtVIT1*’s expression spans both the roots and aboveground parts of plants [15]. Reduced *VITH4* expression signifies a decline in iron sequestration within vesicles [51]. *FRO7* is indispensable for plants to acquire chloroplast iron under iron-limited conditions. Impaired *FRO7* function leads to diminished iron accumulation in chloroplasts, affecting photosynthetic performance and hampering plant growth due to iron deficiency [52]. While *MtFRO4* is expressed solely in the aboveground parts of plants under iron-sufficient conditions, its expression surges in both the aboveground parts and the roots under iron-deficient conditions [53]. In the present study, MT exhibited a downregulatory effect on the relative expression of *VITH4*, *NRAMP3*, and *PG*, while inducing an upregulation of *FRO7*, *FRO4*, *ICS2*, *CCR1*, and *4CL* in peach seedlings under both iron deficiency and iron supply conditions. These findings are consistent with the observations of Lin et al. (2022) [24], suggesting that MT could regulate iron uptake metabolism in peach seedlings. Furthermore, MT plays a role in maintaining iron homeostasis in plants by overseeing iron storage and reutilization mechanisms.

## 4. Materials and Methods

### 4.1. Materials

Wild peach (*Prunus persica*) seeds were collected from a 5-year-old peach tree located on the Chengdu Campus of Sichuan Agricultural University (30°42′ N, 103°51′ E). The nurturing of peach seedlings followed the protocols outlined by Lin et al. (2022) [24] and took place in May 2020. The melatonin employed in the experiment was procured from Beijing Solarbio Science & Technology Co., Ltd. (Beijing, China).

### 4.2. Experimental Design

Upon reaching a height of approximately 8 cm (June 2020), the peach seedlings were transplanted into 50-hole plug trays filled with a mixture of vermiculite and white perlite (1:1, *v*/*v*). Each plug tray accommodated 25 peach seedlings, with a one-hole interval between plantings. The culture substrates were irrigated with deionized water every 3 days to induce iron starvation, continuing until the young leaves of the plants exhibited a yellowish-white hue. Subsequently, the following treatments were conducted: (1) iron supply (Fe), (2) iron deficiency (−Fe), (3) MT application + iron supply (MT+Fe), and (4) MT application + iron deficiency (MT−Fe). Each treatment was replicated three times, utilizing a completely randomized design. Hoagland’s nutrient solution containing 10 mg/L iron (Fe^2+^) was used for the iron supply treatments [54], while the iron deficiency treatments employed a solution devoid of iron. The MT solution (100 µmol/L) was sprayed onto both sides of the leaves and stem surfaces of the peach seedlings for MT treatments, adhering to the methods detailed by Lin et al. (2022) [24]. Approximately 40 mL of the MT solution was applied to each tray. For treatments without MT application, an equivalent volume of deionized water was sprayed. MT solution or deionized water was sprayed every 7 days for a total of four applications. One month after the initial MT spraying, plant samples were collected for subsequent parameter assessments.

### 4.3. Determination of Photosynthetic Pigment Contents

For the −Fe treatment, mature upper leaves (displaying evident green loss) and mature lower leaves (minimal green loss) were harvested to determine the contents of photosynthetic pigments, including chlorophyll *a*, chlorophyll *b*, and carotenoids. Similarly, for the Fe, MT+Fe, and MT−Fe treatments, upper and lower leaves were collected from corresponding positions, as in the −Fe treatment. The photosynthetic pigment contents were assessed using the acetone and ethanol extraction method, as outlined in Xiong (2003) [55].

### 4.4. Extraction of Cell Wall Components and Determination of Cell Wall Component Iron Contents

Fresh samples were utilized for the extraction of cell wall components (cell wall, pectin, and hemicellulose), following the protocols reported by Lei (2014) [56] and Wu et al. (2019) [57]. The quantities of pectin and hemicellulose were determined through the carbazole colorimetric method [58] and the phenol–sulfuric acid method [57], respectively. Contents of cell wall iron, pectin iron, and hemicellulose iron were quantified via iCAP 6300 ICP-MS spectrometry (Thermo Scientific, Waltham, MA, USA), as established by Liu et al. (2001) [59] and Lin et al. (2022) [24].

### 4.5. Contents of Total Iron and Active Iron

Finely ground, dried plant samples were used to assess the contents of total iron and active iron. A mixture of HNO_3_ and HClO_4_ was employed to digest the plant samples for total iron content determination using iCAP 6300 ICP-MS spectrometry. Additionally, another set of plant samples was subjected to extraction with 1 mol L^−1^ HCl (mass-to-volume ratio of 1:10) to gauge the active iron content, also determined through iCAP 6300 ICP-MS spectrometry, as per the methods outlined by Liu et al. (2001) [59] and Lin et al. (2022) [24].

### 4.6. Enzyme Activities: Polygalacturonase (PG), Cellulose (Cx), FCR, and Lignin-Metabolism-Related Enzymes

Equal quantities of fresh upper and lower leaves were combined to extract enzyme solutions for PG and Cx, following the method outlined by Li (2017) [60]. The enzyme solutions were employed to determine PG activity through the colorimetric method, and Cx activity via the 3,5-dinitrosalicylic acid (DNS) method, as per Cao et al. (2007) [58]. For FCR activity assessment, fresh roots were immersed in a CaSO_4_ solution and subsequently introduced into the FCR activity assay reaction solution, following Zuo and Zhang (2004) [61] and Zhao and Ling (2007) [62].

Similarly, equal amounts of fresh upper and lower leaves were mixed to assay the activities of lignin-metabolism-related enzymes, including phenylalanine ammonia-lyase (PAL), cinnamic acid-4-hydroxylase (C4H), cinnamoyl-coenzyme A reductase (CCR), 4-coumarate coenzyme A ligase (4CL), and cinnamyl alcohol dehydrogenase (CAD). These assays employed PAL, C4H, CCR, 4CL, and CAD kits (Nanjing Jiancheng Biological Institute Co., Ltd., Nanjing, China), adhering to the manufacturer’s instructions.

### 4.7. Transcriptome Sequencing and Data Analysis

Mature second and third leaves from the top were collected for transcriptome sequencing. RNA extraction from the leaves was carried out using the RNAprep Pure Plant Kit (Tiangen, Beijing, China). The extracted RNA’s integrity was assessed using the Agilent 2100 (Agilent Technologies, Santa Clara, CA, USA) following the manufacturer’s guidelines. Subsequently, RNA sequencing libraries were generated using the Illumina HiSeq 2500 (Illumina Trading Co., Ltd., Shanghai, China) following the manufacturer’s protocols.

Post-sequencing, the acquired raw data (raw reads) underwent processing, followed by sequence comparison against the peach reference genome. Subsequent differential expression analysis between treatments was performed employing DESeq2, with genes exhibiting an adjusted *p*-value of <0.01 and a fold change of ≥1.5 being identified as differentially expressed. Unigenes were then individually assembled for each sample’s data using Trinity v2.4.0 software (Broad Institute, Cambridge, MA, USA). The final unigene sequences were integrated with the COG, GO, KEGG, KOG, NR, Pfam, Swiss-Prot, and EggNOG databases, and unigene annotation information was assessed through comparison using Blast 2.9.0 software (National Library of Medicine, Bethesda, MD, USA). Enrichment analyses for gene ontology (GO) and the Kyoto Encyclopedia of Genes and Genomes (KEGG) were carried out for the differentially expressed genes (DEGs).

### 4.8. Quantitative RT-PCR

The relative expression levels of selected DEGs were determined through quantitative RT-PCR (qRT-PCR). Primers were designed using Primer 5.0 software, based on reference gene sequences in peach (Table 4). *TEF2* served as the internal reference gene following Zhang et al. (2014) [30]. The designed primers were synthesized by Tsingke Biotechnology Co., Ltd. (Chengdu, China). qRT-PCR of the DEGs was performed on the CFX96TM Real-Time System platform, utilizing the 2X M5 HiPer SYBR Premix EsTaq (with Tli RNaseH) kit from Mei5 Biotechnology Co., Ltd. (Beijing, China). Relative DEG expression levels were calculated employing the 2^−ΔΔ*C*T^ method [63].

### 4.9. Statistical Analysis

Statistical analyses were conducted using SPSS 20.0 (IBM, Inc., Armonk, NY, USA). The data were subjected to one-way analysis of variance (ANOVA), followed by Duncan’s multiple range test (*p* < 0.05).

## 5. Conclusions

Under iron deficiency, the application of MT demonstrated several key effects on peach plants. MT exhibited the capability to enhance the content of photosynthetic pigments and finely regulate the activities of PG, Cx, PAL, CCR, and C4H. This regulatory activity extended to the reactivation of cell wall iron, pectin iron, and hemicellulose iron within the plant’s cellular structures. Furthermore, MT intervention effectively elevated the activity of FCR and increased the contents of total and active iron in various organs in response to iron deficiency. Moreover, MT regulated the aminoacyl-tRNA biosynthesis pathway. Mechanistically, MT downregulated the expression levels of *VITH4*, *NRAMP3*, and *PG*, while simultaneously upregulating the expression of *FRO7*, *FRO4*, *ICS2*, *CCR1*, and *4CL*. These regulations collectively contributed to maintaining iron homeostasis within peach plants by promoting efficient iron storage and reutilization mechanisms under iron-deficient conditions. Through these findings, this study significantly advances our understanding of the intricate molecular mechanisms underlying MT-mediated regulation of iron reactivation and reutilization within peach plants confronting iron deficiency conditions.

## Figures and Tables

**Figure 1 ijms-24-16133-f001:**
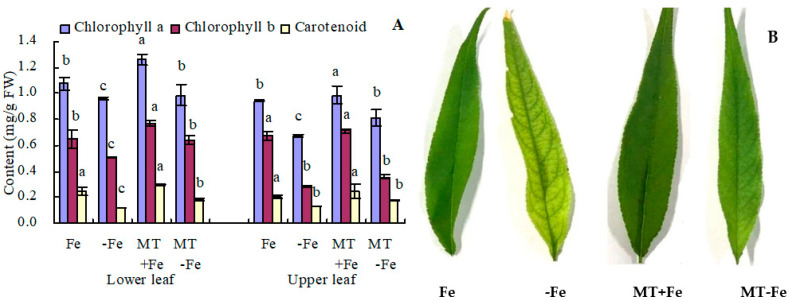
Photosynthetic pigment content (**A**) and upper leaf color (**B**) of peach seedlings. Values are means (±SE) of three replicates. Different letters indicate significant differences among the treatments (Duncan’s multiple range test, *p* < 0.05). FW = fresh weight.

**Figure 2 ijms-24-16133-f002:**
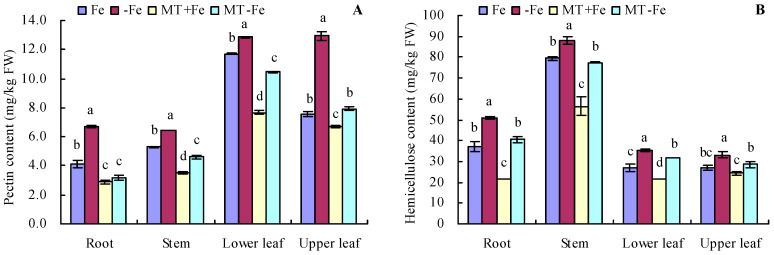
Contents of pectin (**A**) and hemicellulose (**B**) in peach seedlings. Values are means (±SE) of three replicates. Different letters indicate significant differences among the treatments (Duncan’s multiple range test, *p* < 0.05). FW = fresh weight.

**Figure 3 ijms-24-16133-f003:**
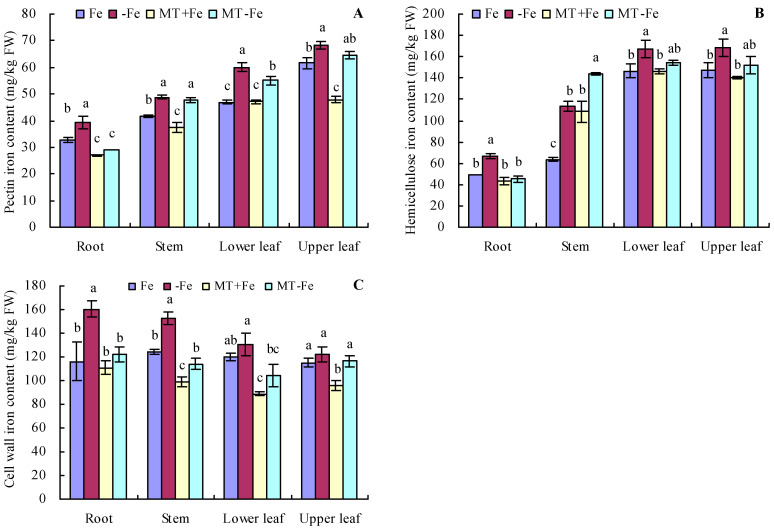
Contents of pectin iron (**A**), hemicellulose iron (**B**), and cell wall iron (**C**) in peach seedlings. Values are means (±SE) of three replicates. Different letters indicate significant differences among the treatments (Duncan’s multiple range test, *p* < 0.05). FW = fresh weight.

**Figure 4 ijms-24-16133-f004:**
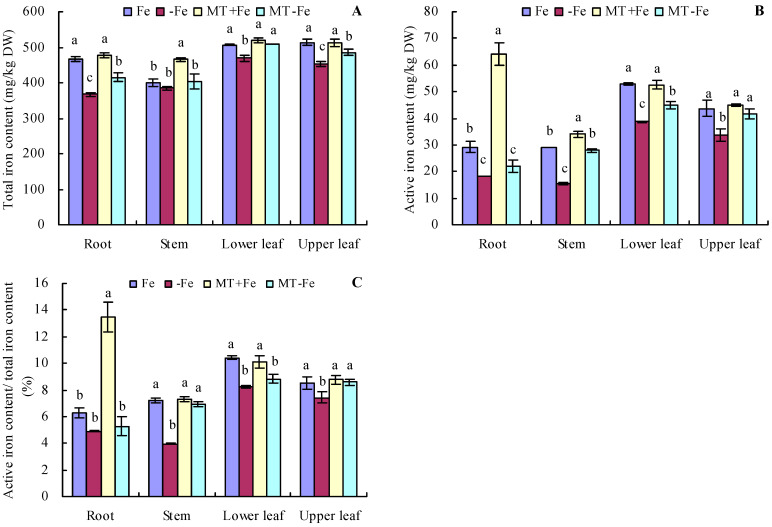
Contents of total iron (**A**), active iron (**B**), and the proportion of active iron content to total iron content (**C**) in peach seedlings. Values are means (±SE) of three replicates. Different letters indicate significant differences among the treatments (Duncan’s multiple range test, *p* < 0.05). DW = dry weight.

**Figure 5 ijms-24-16133-f005:**
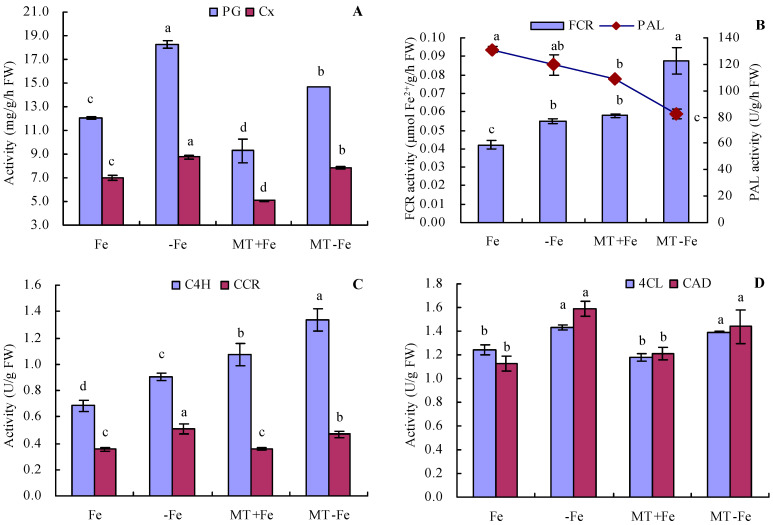
Activities of PG, Cx, FCR, and lignin-metabolism-related enzyme in peach seedlings. (**A**): activities of polygalacturonase (PG) and cellulase (Cx); (**B**): activities of ferric-chelate reductase (FCR) and phenylalanine ammonia-lyase (PAL); (**C**): activities of cinnamic acid-4-hydroxylase (C4H) and cinnamoyl-coenzyme A reductase (CCR); (**D**): activities of 4-coumarate coenzyme A ligase (4CL) and cinnamyl alcohol dehydrogenase (CAD). Values are means (±SE) of three replicates. Different letters indicate significant differences among the treatments (Duncan’s multiple range test, *p* < 0.05). FW = fresh weight.

**Figure 6 ijms-24-16133-f006:**
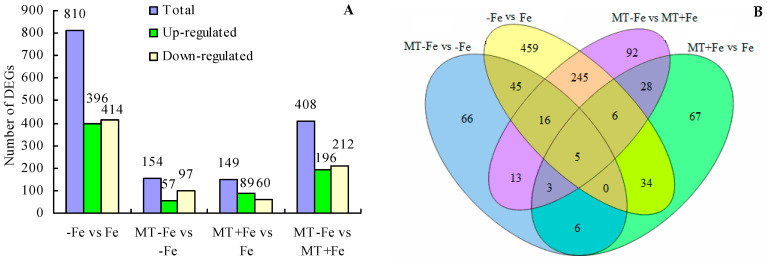
Statistical diagram of DEGs. (**A**) Number of DEGs; (**B**) Venn diagram of DEGs.

**Figure 7 ijms-24-16133-f007:**
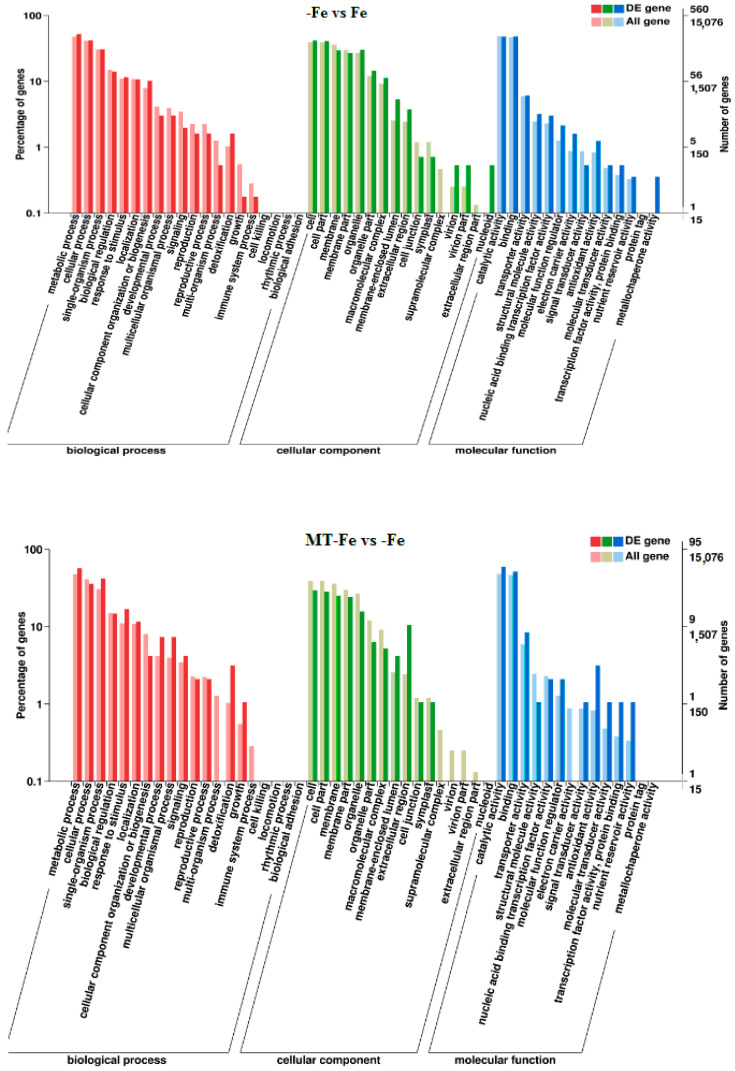
GO enrichment analysis of DEGs.

**Figure 8 ijms-24-16133-f008:**
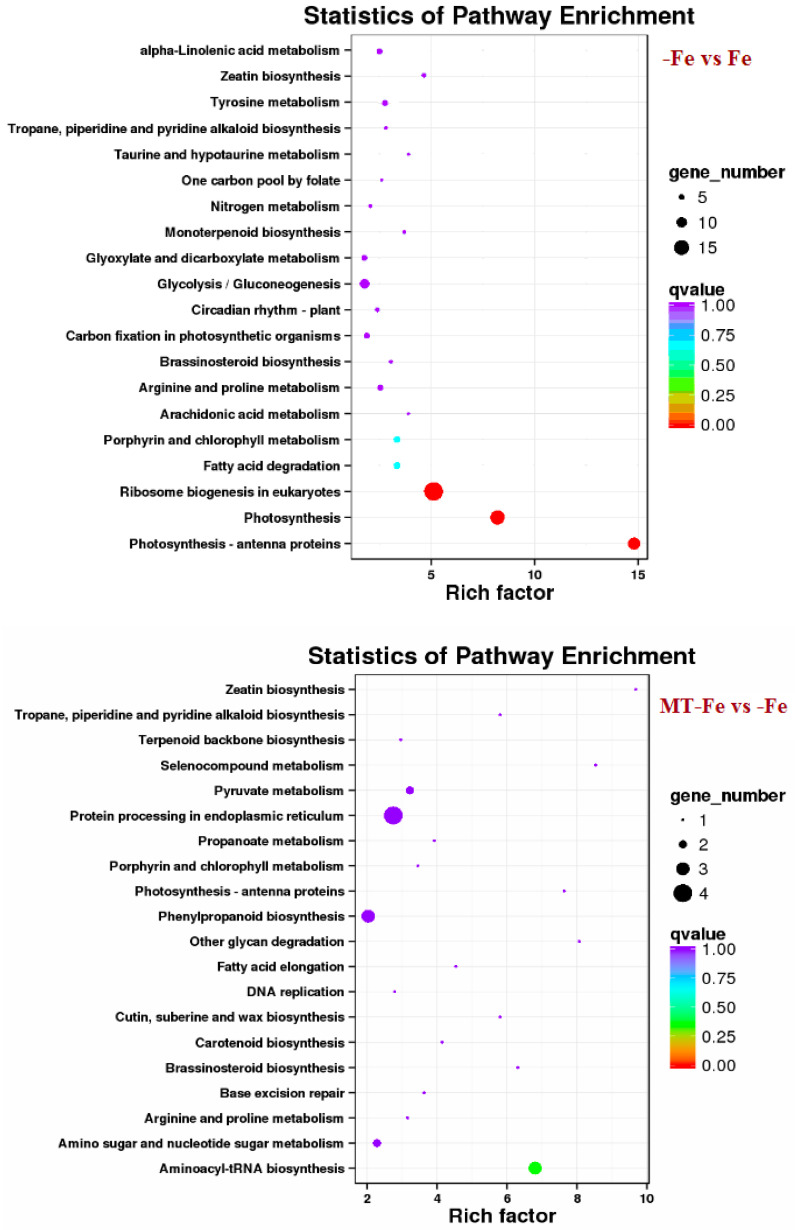
KEGG enrichment analysis of DEGs.

**Figure 9 ijms-24-16133-f009:**
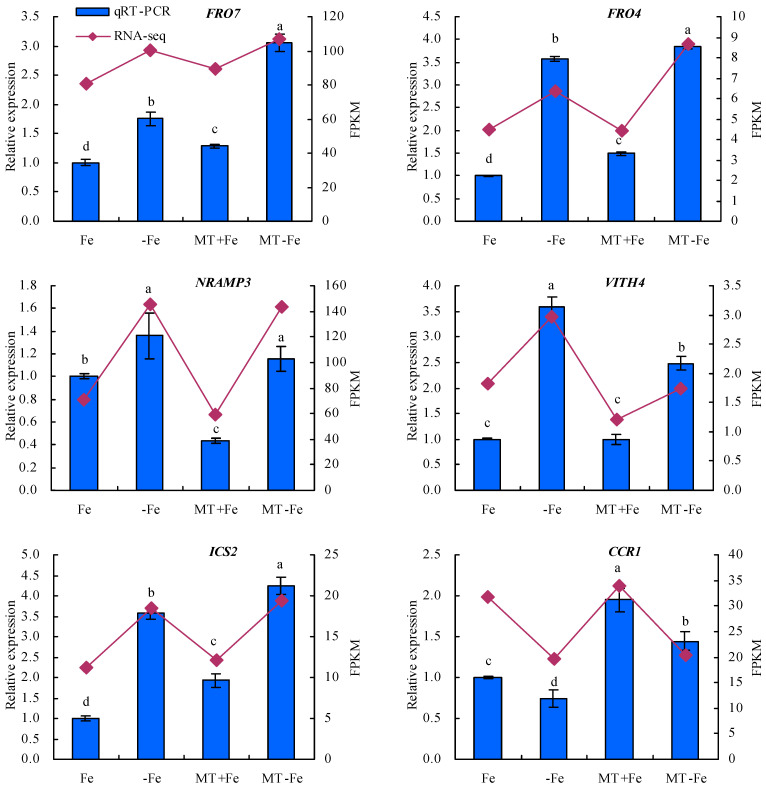
Relative expressions of screened DEGs. Values are means (±SE) of three replicates. Different letters indicate significant differences among the treatments (Duncan’s multiple range test, *p* < 0.05).

**Table 1 ijms-24-16133-t001:** Statistics on the number of annotated DEGs.

DEG Set	Total	COG	GO	KEGG	KOG	NR	Pfam	Swiss-Prot	EggNOG
−Fe vs. Fe	806	356	560	338	437	806	693	637	764
MT−Fe vs. −Fe	154	57	95	49	82	154	119	114	136
MT+Fe vs. Fe	147	68	81	57	80	147	118	118	130
MT−Fe vs. MT+Fe	406	184	282	153	192	406	348	321	375

**Table 2 ijms-24-16133-t002:** KEGG classification of DEGs.

DEG Set	KEGG Pathway	KO ID	Gene Number	Upregulated	Downregulated	*p*-Value
−Fe vs. Fe	Photosynthesis antenna proteins	ko00196	12	0	12	<0.001
	Photosynthesis	ko00195	14	13	1	<0.001
	Ribosome biogenesis in eukaryotes	ko03008	19	0	19	<0.001
	Fatty acid degradation	ko00071	6	4	2	0.008
	Porphyrin and chlorophyll metabolism	ko00860	6	3	3	0.008
	Tyrosine metabolism	ko00350	6	3	3	0.017
	Zeatin biosynthesis	ko00908	3	2	1	0.024
	Alpha-linolenic acid metabolism	ko00592	5	3	2	0.045
	Arginine and proline metabolism	ko00330	5	4	1	0.045
	Monoterpenoid biosynthesis	ko00902	3	2	1	0.045
MT−Fe vs. −Fe	Aminoacyl-tRNA biosynthesis	ko00970	3	1	2	0.009
MT+Fe vs. Fe	Protein processing in endoplasmic reticulum	ko04141	11	10	1	<0.001
	Monoterpenoid biosynthesis	ko00902	3	2	1	0.001
MT−Fe vs. MT+Fe	Photosynthesis antenna proteins	ko00196	6	6	0	<0.001
	Photosynthesis	ko00195	5	5	0	0.001
	Ubiquinone and other terpenoid-quinone biosynthesis	ko00130	4	3	1	0.009
	Porphyrin and chlorophyll metabolism	ko00860	4	2	2	0.012
	Tyrosine metabolism	ko00350	4	3	1	0.020
	Taurine and hypotaurine metabolism	ko00430	2	1	1	0.026
	Glycolysis/Gluconeogenesis	ko00010	6	4	2	0.039

**Table 3 ijms-24-16133-t003:** Transcription factor analysis of DEGs.

Gene ID	*p* Value	log_2_FC	Transcription Factor Families	NR_Annotation
gene3903	9.97 × 10^−3^	−0.58797	Trihelix	Trihelix transcription factor GTL2
gene12799	1.56 × 10^−4^	−0.58876	TCP	Transcription factor TCP2
gene19160	2.44 × 10^−9^	−0.69178	RAP	Ethylene-responsive transcription factor RAP2-4
gene24513	8.44 × 10^−6^	1.07499	RADIALIS	Transcription factor RADIALIS
gene20320	8.84 × 10^−12^	−0.83518	PIF	Transcription factor PIF3 isoform X1
gene13809	3.01 × 10^−3^	−0.84145	NAC	NAC-domain-containing protein 72
gene4334	4.93 × 10^−4^	−0.59185	MYB	Transcription factor MYB3R-1
gene3908	9.31 × 10^−3^	−1.06674	MYB	Transcription factor MYB6
gene2530	7.35 × 10^−6^	0.73483	MYB	Transcription factor MYB1R1
gene22870	1.67 × 10^−6^	−1.06405	MYB	Transcription factor MYB6
gene13985	5.16 × 10^−4^	1.05636	MYB	Transcription factor TT2 isoform X1
gene10551	9.54 × 10^−5^	1.89694	MYB	Transcription factor MYB114
gene13869	6.65 × 10^−3^	0.70715	IFH	Transcriptional regulator IFH1
gene8214	5.76 × 10^−5^	0.70881	ICE1	Transcription factor ICE1 isoform X1
gene4595	1.99 × 10^−7^	0.59439	HY5	Transcription factor HY5
gene8513	8.85 × 10^−4^	−0.95029	HSP	Heat stress transcription factor B-2a
gene3957	4.67 × 10^−3^	0.78700	HSP	Heat shock factor protein HSF30 isoform X2
gene23145	2.01 × 10^−5^	−0.69709	HSP	Heat stress transcription factor C-1
gene22233	4.95 × 10^−3^	0.71842	HSP	Heat stress transcription factor B-1
gene18345	7.06 × 10^−3^	−0.82542	GATA	GATA transcription factor 11 isoform X1
gene343	2.12 × 10^−3^	1.38468	ERF	Ethylene-responsive transcription factor 1B [*Prunus avium*]
gene16074	2.27 × 10^−5^	−0.68557	ERF	Ethylene-responsive transcription factor CRF2
gene15540	6.00 × 10^−3^	−0.96760	ERF	Ethylene-responsive transcription factor 1A
gene4875	7.77 × 10^−3^	0.67635	bZIP	Transcription factor TGA7 isoform X1
gene16798	1.07 × 10^−20^	2.52974	bHLH	Transcription factor bHLH47
gene15277	7.79 × 10^−5^	−1.56733	bHLH	Transcription factor bHLH71
gene12740	1.22 × 10^−5^	1.00272	bHLH	Transcription factor bHLH144 isoform X1
gene16285	6.39 × 10^−3^	−0.90251	AP2	AP2-like ethylene-responsive transcription factor PLT1
gene8614	4.17 × 10^−6^	−2.08404	WRKY	Low-quality protein: probable WRKY transcription factor 27
gene8260	5.83 × 10^−3^	−1.20454	WRKY	Probable WRKY transcription factor 70
gene3867	1.09 × 10^−9^	−0.70795	WRKY	Probable WRKY transcription factor protein 1 isoform X3
gene26136	4.64 × 10^−6^	−0.66771	WRKY	Probable WRKY transcription factor 69 isoform X1
gene20240	4.82 × 10^−3^	−1.05808	WRKY	Probable WRKY transcription factor 70

**Table 4 ijms-24-16133-t004:** Primers for qRT-PCR.

No.	Gene Name	Gene ID in NCBI	F (Sequence 5′-3′)	R (Sequence 5′-3′)
1	*FRO7*	835037	TTTCACAATGGCTGCTGGAGGA	CACATGGAAGGCACTTCGCTGA
2	*FRO4*	832463	AGGCTCCTCTGGGAATTGTTAC	TCATACACTTTCTCGCCATCTT
3	*NRAMP3*	816847	ATCTTCTGCTGGATTTCTTCTC	TTGAGGTTGTGGCAATTACACT
4	*VITH4*	25493306	CAAACGACCTAGAACACCAACA	TGAGCCACCTCTATGTCCAACT
5	*ICS2*	838468	TGACCAGATTCAATCGGAACAC	TAAGTGCGAATAAGCGGACATT
6	*CCR1*	838165	TGACTAATGACAAGCCCTACCT	CTTTCCCATAGCAGTACCAGTT
7	*PG*	102578026	TTGTTGGAATGCTTATGGGACT	AGATAAATGGCTCTTGGGCTCT
8	*4CL*	100245991	GCCAGTGATTAAGCAGCAAGAC	GCGACAACCCGTAGATATGAAA
9	*TEF2*	733027	GGTGTGACGATGAAGAGTGATG	TGAAGGAGAGGGAAGGTGAAAG

## Data Availability

The data presented in this study are available upon request from the corresponding author. The raw sequence data reported in this paper have been deposited in the National Center for Biotechnology Information (NCBI) and are publicly accessible at https://www.ncbi.nlm.nih.gov/sra/PRJNA1017989 (accessed on 16 September 2023).

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
