# Peer review of "Melatonin Promotes Iron Reactivation and Reutilization in Peach Plants under Iron Deficiency"

_ijms, 2023, doi:10.3390/ijms242216133_

Round 1

Reviewer 1 Report

Comments and Suggestions for Authors

 ‘Melatonin Promotes Iron Reactivation and Reutilization of Peach Plants under Iron Deficiency’ by Lin et al. 2021 reports that MT enhances the content of photosynthetic pigments, regulates the activity of several enzymes, including FCR, Fe content in different tissues, and regulates gene expression. While the authors conducted several studies to understand the mechanism of MT in Fe homeostasis, the manuscript has some limitations.

MT has been shown to alleviate Fe deficiency stress in several plant species, including apples, cucumbers, and peaches (same group). MT is also reported to alleviate other abiotic stress, such as cold; therefore, the originality of the manuscript is questionable.

In Figure 4A, MT increases FCR activity in +Fe more strongly than the -Fe or -Fe+MT. It is not clear why FCR activity is enhanced by MT than -Fe alone. Also, this finding contradicts Figure 8, which shows that the expression of FRO genes, which encode the FCR, decreased with Fe with or without MT. This data needs to be vaidated. 

'NRAMP functions in plants as a central transporter for iron absorption and transportation'; this statement is not accurate. NRMAPs may transport Fe, but they are not specific to Fe and they also mediate the transport of other metals such as Mn.  

Figures and duplicated and hard to follow. Figure label fonts are also too small to understand.

Comments on the Quality of English Language

The English language requires extensive editing.

Author Response

‘Melatonin Promotes Iron Reactivation and Reutilization of Peach Plants under Iron Deficiency’ by Lin et al. 2021 reports that MT enhances the content of photosynthetic pigments, regulates the activity of several enzymes, including FCR, Fe content in different tissues, and regulates gene expression. While the authors conducted several studies to understand the mechanism of MT in Fe homeostasis, the manuscript has some limitations.

RESPONSE: Thank you for your reviewing.

MT has been shown to alleviate Fe deficiency stress in several plant species, including apples, cucumbers, and peaches (same group). MT is also reported to alleviate other abiotic stress, such as cold; therefore, the originality of the manuscript is questionable.

RESPONSE: In the previous studies, MT has been reported to alleviate the abiotic stress, such as cold, high temperature, heavy metal. Under normal condition, MT can promote the Fe uptakes in grapes and peaches. There is only the study of MT alleviating Fe deficiency stress for cucumbers, and no study of Fe reactivation and reutilization. This experiment is a further study on the effects of exogenous MT on iron absorption and transport mechanisms within peach plants under iron-deficient conditions. So, we think the originality of the manuscript is no questionable.

In Figure 4A, MT increases FCR activity in +Fe more strongly than the -Fe or -Fe+MT. It is not clear why FCR activity is enhanced by MT than -Fe alone. Also, this finding contradicts Figure 8, which shows that the expression of FRO genes, which encode the FCR, decreased with Fe with or without MT. This data needs to be vaidated.

RESPONSE: We have checked the data. The data was laid out in the wrong order for the graph. We have revised.

'NRAMP functions in plants as a central transporter for iron absorption and transportation'; this statement is not accurate. NRMAPs may transport Fe, but they are not specific to Fe and they also mediate the transport of other metals such as Mn.

RESPONSE: We have revised as “NRAMP can regulate the iron absorption and transportation.”

Figures and duplicated and hard to follow. Figure label fonts are also too small to understand.

RESPONSE: We have deleted the duplicated Figures, and made the label fonts bigger.

In addition, the English language have been edited.

Reviewer 2 Report

Comments and Suggestions for Authors

ijms-2622596

Major comments:

The iron content per cell wall weight, the proportion of cell wall iron to total iron in Figure 2, and the proportion of active iron to total iron in Figure 3 are necessary to discuss the effects of MT on the reutilization of iron. Organs of peach seedlings analyzed in Figure 2 and 3 should also be specified in the figure titles to discuss the re-allocation of iron in plants. Please improve the third paragraph of the Discussion section based on the additional information mentioned above.

The literatures cited in the first paragraph of the Introduction section do not meet the requirements. Review papers or chapters of the well-known textbook concerning general iron nutrition of plants are relevant for the first sentence, and those concerning the chemistry of iron and chelation are relevant for the second sentence. Since the authors claimed the reutilization of iron bound to the cell wall, the chemistry of iron is important for discussion. Multiple literatures are necessary for the third sentence because the authors did not restrict the description to a specific plant species. Related to this matter, the first part of the Discussion (Iron plays … MT curtails chlorophyll degradation in plants.) is suitable for the Introduction.

Minor comments:

Abstract ‘MT led to heightened levels of …’

Please clarify the entire plant body or specific organs.

2.1. ‘Wild peach’

‘Wild peach’ should be defined by scientific name. ‘Wild’ means ‘not breeding’ or ‘volunteer plant’?

3.1. ‘MT treatment deepened the color intensity …’

The reason why the MT treatment deepened the color intensity should be described. Accumulation of anthocyanins also deepens the leaf color. Why do authors present only pictures, but not the values of pigments for the middle leaves?

Discussion, the first paragraph

‘normal plant physiology’ is vague. Please concretely describe.

Discussion, the last paragraph

The line break is needless.

Literature 15

Is this the thesis? Bibliographic information is necessary to access the body of literature.

Author Response

The iron content per cell wall weight, the proportion of cell wall iron to total iron in Figure 2, and the proportion of active iron to total iron in Figure 3 are necessary to discuss the effects of MT on the reutilization of iron. Organs of peach seedlings analyzed in Figure 2 and 3 should also be specified in the figure titles to discuss the re-allocation of iron in plants. Please improve the third paragraph of the Discussion section based on the additional information mentioned above.

RESPONSE: Thank you for your reviewing. We have added the proportion of active iron to total iron in Figure 3 and Discussion. Because no determination data cell wall weight, it can’t calculate the iron content per cell wall weight. In addition, the determination of cell wall iron content was used fresh samples, while the determination of total iron content was used dry samples. So, it also can’t calculate the proportion of cell wall iron to total iron.

The literatures cited in the first paragraph of the Introduction section do not meet the requirements. Review papers or chapters of the well-known textbook concerning general iron nutrition of plants are relevant for the first sentence, and those concerning the chemistry of iron and chelation are relevant for the second sentence. Since the authors claimed the reutilization of iron bound to the cell wall, the chemistry of iron is important for discussion. Multiple literatures are necessary for the third sentence because the authors did not restrict the description to a specific plant species. Related to this matter, the first part of the Discussion (Iron plays … MT curtails chlorophyll degradation in plants.) is suitable for the Introduction.

RESPONSE: We have rewritten the first paragraph of Introduction, and moved the first part of the Discussion to Introduction. The chemistry of iron is introduced in second paragraph of Introduction.

Minor comments:

Abstract ‘MT led to heightened levels of …’

Please clarify the entire plant body or specific organs.

RESPONSE: We have revised as “MT led to heightened levels of photosynthetic pigments (chlorophyll a, chlorophyll b, and carotenoids) in peach leaves and decreased concentrations of pectin, hemicellulose, cell wall iron, pectin iron, and hemicellulose iron in peach plants”.

2.1. ‘Wild peach’

‘Wild peach’ should be defined by scientific name. ‘Wild’ means ‘not breeding’ or ‘volunteer plant’?

RESPONSE: It should be Prunus persica. We have added in the text.

3.1. ‘MT treatment deepened the color intensity …’

The reason why the MT treatment deepened the color intensity should be described. Accumulation of anthocyanins also deepens the leaf color. Why do authors present only pictures, but not the values of pigments for the middle leaves?

RESPONSE: MT treatment just deepened the green color intensity. The reason is because of MT increasing the chlorophyll a content. Accumulation of anthocyanins deepens the other color intensity. It should be the middle leaves of upper part. We have revised as “upper leaves.”

Discussion, the first paragraph

‘normal plant physiology’ is vague. Please concretely describe.

RESPONSE: It should be “photosynthetic physiology”. We have revised.

Discussion, the last paragraph

The line break is needless.

RESPONSE: We have revised.

Literature 15

Is this the thesis? Bibliographic information is necessary to access the body of literature.

RESPONSE: It is the master thesis. We have checked and revised the literature 15.

Round 2

Reviewer 2 Report

Comments and Suggestions for Authors

The manuscript was satisfactorily revised.

L61, ‘nicotinamide’ à ‘nicotianamine’

L116, because of

This section is Results, not Discussion. ‘consistent’ or ‘not contradict’ are suitable words.

Author Response

The manuscript was satisfactorily revised.

Thank you for your reviewing.

L61, ‘nicotinamide’ à ‘nicotianamine’

We have revised as ‘nicotianamine’.

L116, because of

This section is Results, not Discussion. ‘consistent’ or ‘not contradict’ are suitable words.

We have revised as “which was consistent with the increase of chlorophyll a content by MT”.